

# Meta-analysis of the efficacy of the erector spinae plane block after spinal fusion surgery

Yi He[1], Heng Liu[1], Peng Ma[2], Jing Zhang[3] and Qiulian He[4]

[1] Department of Sports Medicine, Nanbu People's Hospital (Spine, Upper Limb Orthopedics, Sports Medicine), Nanchong, China
[2] Department of Otorhinolaryngology Head and Neck Surgery, Nanchong Central Hospital, Beijing Anzhen Hospital, Capital Medical University, Nanchong, China
[3] Medical Department of Nanbu County People's Hospital, Nanchong, China
[4] Department of Hematology, Nanchong Central Hospital, Beijing Anzhen Hospital, Capital Medical University, Nanchong, China

## ABSTRACT

**Objective**. To investigate the efficacy of erector spinal plane block (ESPB) after spinal fusion surgery in this study.

**Methods**. The PubMed, Embase, Cochrane library, and Web of Science databases were searched with a search deadline of March 30, 2024, and Stata 15.0 was used to analyze the data from the included studies.

**Result**. Nine randomized controlled trials involving 663 patients were included. Meta-analysis showed that EPSB could reduce pain scores at 2h (standard mean difference (SMD) = $-0.78$, 95% CI [$-1.38$ to $-0.19$], GRADE: Moderate), 6 h (SMD = $-0.81$, 95% CI [$-1.23$ to $-0.38$], GRADE: Moderate), 12 h (SMD = $-0.59$, 95% CI [$-1.05$ to $-0.13$], GRADE: Moderate), 24 h (SMD = $-0.54$, 95% CI [$-0.86$ to $-0.21$], GRADE: Moderate), 48 h (SMD = $-0.40$, 95% CI [$-0.75$ to $-0.05$], GRADE: Moderate) after spinal fusion surgery, as well as the PCA (analgesia medication use) (SMD = $-1.67$, 95% CI [$-2.67$ to $-0.67$], GRADE: Moderate). However, EPSB had no effect on intraoperative blood loss (SMD = $-0.28$, 95% CI [$-1.03$ to 0.47], GRADE: Low) and length of hospital stay (SMD = $-0.27$, 95% CI [$-0.60$–0.06], GRADE: Low).

**Conclusion**. Combined with the current findings, EPSB may reduce pain scores in spinal fusion surgery, possibly reducing the use of postoperative analgesics. However, due to the limitations of the study, we need more high-quality, multi-center, large sample randomized controlled trials to merge.

## INTRODUCTION

In recent years, the incidence of spinal diseases has been increasing year by year, and the number of spinal fusion surgeries has also increased (*Bonfiglio et al., 2021*), spinal fusion is a procedure in which two or more vertebrae are permanently joined together in order to eliminate movement between the vertebrae in order to stabilize the spine and reduce pain. This procedure differs from other spinal surgeries in that it uses bone grafts and internal fixation devices to achieve fusion of the vertebrae, thereby limiting the flexibility

Corresponding author
Qiulian He, 38299320@qq.com

of the spine. This procedure is often used to treat spinal degenerative diseases, spinal instability, and other problems, but may result in degeneration of adjacent segments and reduced range of motion. However, the unique structure of the spine makes spinal fusion surgery very risky, and spinal fusion surgery usually cause severe pain, which is easy to cause sympathetic excitation of the patient and increase the sensitivity to pain, and the use of opioids in such surgeries has a high dose in the perioperative period, and the postoperative development of narcosis has a high risk (*Stewart et al., 2024*; *Tulgar, Ermis & Ozer, 2018*). The use of opioid analgesics during the perioperative period for this type of surgery is associated with higher risk of adverse effects such as drowsiness, hypotension, bradycardia, respiratory depression, nausea, and vomiting. Severe postoperative pain can lead to delayed mobility, deep vein thrombosis, delirium, and chronic pain syndrome (*Neuman, Sieber & Dillane, 2023*; *Tang et al., 2022*). Erector spinae plane block (ESPB) is a new method of interfacial plane block (*Chin, 2019*; *Helander et al., 2019*). In 2016, *Forero et al. (2016)* first published the analgesic application of ESPB in thoracic neuropathic pain, describing ESPB as a simple, effective, and safe technique for chronic neuropathic pain in the chest and acute postoperative or post-traumatic pain in the chest. It is described that ESPB is a simple, effective, and safe technique for chronic neuropathic pain in the chest and acute postoperative or posttraumatic pain in the chest (*Oostvogels et al., 2024*; *Schnabel et al., 2023*). In the past two years, ultrasound guided ESPB technology has been gradually applied to thoracic, abdominal, gynecological, and various kinds of laparoscopic surgeries and postoperative analgesia in foreign countries, and the clinical effect is remarkable (*Sørenstua, Leonardsen & Chin, 2024*; *Tsui et al., 2019*). ESPB is usually combined with general analgesia for intraoperative and postoperative analgesia and can also be used alone for some surgeries with little pain stimulation. ESPB can effectively alleviate the postoperative pain of the patients, promote the recovery of the patients, minimize the occurrence of complications, and show obvious advantages (*Coppens et al., 2023*). Therefore, EPSB can be used as analgesia for spinal fusion surgery, but since the effect of EPSB for spinal fusion surgery is still controversial (*Canturk, 2019*; *Tulgar et al., 2019*). This study combined with the latest research to explore the efficacy of EPSB in spinal surgery, hoping to provide a new choice for spinal fusion surgery patients.

## MATERIALS AND METHODS

This study was conducted based on Prisma checklist reports (*Page et al., 2021*) and was pre-registered on PROSPERO under the registration number CRD42024525373.

### Literature search

The PubMed, Embase, Cochrane library, and Web of Science databases were searched with a March 30, 2024, search deadline. The search was performed using subject and free words with the search terms spinal fusion, erector spinae plane blocks. The specific search strategy is described in Table S1.

### Inclusion criteria

The population included in this study was adults eligible for spinal fusion surgery, with ESPB in the experimental group and conventional treatment in the control group, with

the primary outcome was pain scores, and the secondary outcomes were PCA (analgesia medication use), intraoperative blood loss, and length of hospital stay. This study mainly included randomized controlled studies.

## Exclusion criteria
The following exclusion criteria were considered for this study: 1: Types of literature included (conference abstracts, reviews, meta-analyses, protocols, case reports) 2: Articles for which full text was not available as well as articles for which no usable data was available.

## Data extraction
The literature was meticulously sieved through by two authors (HY and LH) in adherence to stringent inclusion and exclusion criteria. Any disparities were meticulously ironed out through negotiations or by soliciting input from third-party arbiters (HQL), culminating in unanimous accord. Extracted from the incorporated studies are pivotal particulars, encompassing Authorship, publication year, geographical location, sample dimensions, gender distribution, age demographics, intervention modalities, and outcome.

## Risk of bias assessment
The two researchers (HY and LH) independently utilized the Cochrane Collaboration tool (*Higgins et al., 2011*) to assess the risk of bias as low, unclear, or high. In cases of disagreement, a third person (HQL) was consulted for consensus. The evaluation encompassed seven facets: generation of random sequence (selection bias), allocation concealment (selection bias), blinding of implementers and participants (performance bias), blinding of outcome assessors (detection bias), completeness of outcome data (attrition bias), selective reporting of study results (reporting bias), and other potential sources of bias. Each included study was individually assessed based on these criteria. If a study fully met all criteria, its bias was deemed "low risk", indicating high study quality and overall low bias risk. If a study partially met the criteria, its quality was classified as "unclear risk", signifying a moderate likelihood of bias. If a study completely failed to meet the criteria, it was categorized as "high risk", indicating a high risk of bias and low study quality.

## Grade of evidence
To determine the quality of our results, we selected the Graded Recommendations Assessment Development and Evaluation (GRADE) system to evaluate the evidence (*Atkins et al., 2004*) for methodological quality. We considered five factors that could reduce the quality of the evidence, including study limitations, inconsistent findings, inconclusive direct evidence, inaccurate or wide confidence intervals, and publication bias. In addition, three factors that could reduce the quality of evidence were reviewed, namely effect size, possible confounding factors, and dose–effect relationships. A comprehensive description of the quality of evidence for each parameter data is provided. The quality of evidence was divided into high, moderate, low and very low quality.

## Data analysis

Statistical analysis was performed using Stata 15.0 software (Stata Corp, College Station, TX, USA). Heterogeneity among the included studies was assessed using $I^2$ values or the Q statistic. $I^2$ values of 0%, 25%, 50%, and 75% indicated no, low, medium, and high heterogeneity, respectively. When $I^2$ was ≥50%, a random-effects model was used, and sensitivity analyses were performed to explore sources of heterogeneity. If heterogeneity was less than 50%, a fixed-effects model was used for analysis. All outcomes were continuous variables and were combined using SMD and 95% CI. Egger's test was used to assess publication bias, and $P < 0.05$ was the absence of publication bias, while the opposite represented the presence of publication bias.

# RESULT

## Literature screening

An initial search of 82 articles was conducted for this study, and after removing duplicates, reading titles and abstracts, and reading the full text, nine randomized controlled studies were included, and the search flowchart is shown in Fig. 1.

## Basic characteristics and risk of bias of the included literature

Nine randomized controlled studies (*Bellantonio et al., 2023*; *Beltrame, Fasano & Jalón, 2023*; *Gişi & Öksüz, 2023*; *Goel et al., 2021*; *Kumar et al., 2024*; *Nashibi et al., 2023*; *Wang et al., 2021*; *Zhang et al., 2021*; *Zhang et al., 2023*) involving 663 patients were included, published in 2021–2024, with an age range of 50.06–61 years, are shown in Table 1 for their basic characteristics. The all included studies clearly accounted for the method of random assignment used and were therefore evaluated to be low risk, but two studies (*Wang et al., 2021*; *Zhang et al., 2023*) did not account for the blinding method used and were therefore evaluated to be high risk, and the evaluation of risk of bias is shown in Fig. S2. The results of the GRADE ratings for this study are presented in Table S2.

## Results of meta-analysis
### Pain scores

In the analysis of postoperative pain scores at different time points after spinal fusion, epidural paraspinal block (EPSB) was effective in reducing pain across 2-hour, 6-hour, 12-hour, 24-hour, and 48-hour intervals. Specifically, at 2 h postoperatively, a combined analysis of five studies showed that EPSB significantly reduced pain scores (SMD = −0.78, 95% CI [−1.38 to −0.19], $I^2 = 81.9\%$, $P = 0.001$), with heterogeneity potentially originating from *Bellantonio et al. (2023)* (Fig. 2). At 6 h, an analysis of six studies indicated a significant reduction in pain scores (SMD = −0.81, 95% CI [−1.23 to −0.38], $I^2 = 69.7\%$, $P = 0.006$), with stable results (Fig. 3). At 12 h, five studies also demonstrated a significant reduction in pain scores (SMD = −0.59, 95% CI [−1.05 to −0.13], $I^2 = 70.2\%$, $P = 0.009$) (Fig. 4). At 24 h, the combined analysis of six studies showed a decrease in pain scores (SMD = −0.54, 95% CI [−0.86 to −0.21], $I^2 = 50.7\%$, $P = 0.071$) (Fig. 5). Finally, at 48 h, the analysis of two studies indicated a reduction in pain scores (SMD = −0.40, 95% CI [−0.75 to −0.05], $I^2 = 39.9\%$, $P = 0.197$) (Fig. 6). Overall, despite varying levels of heterogeneity
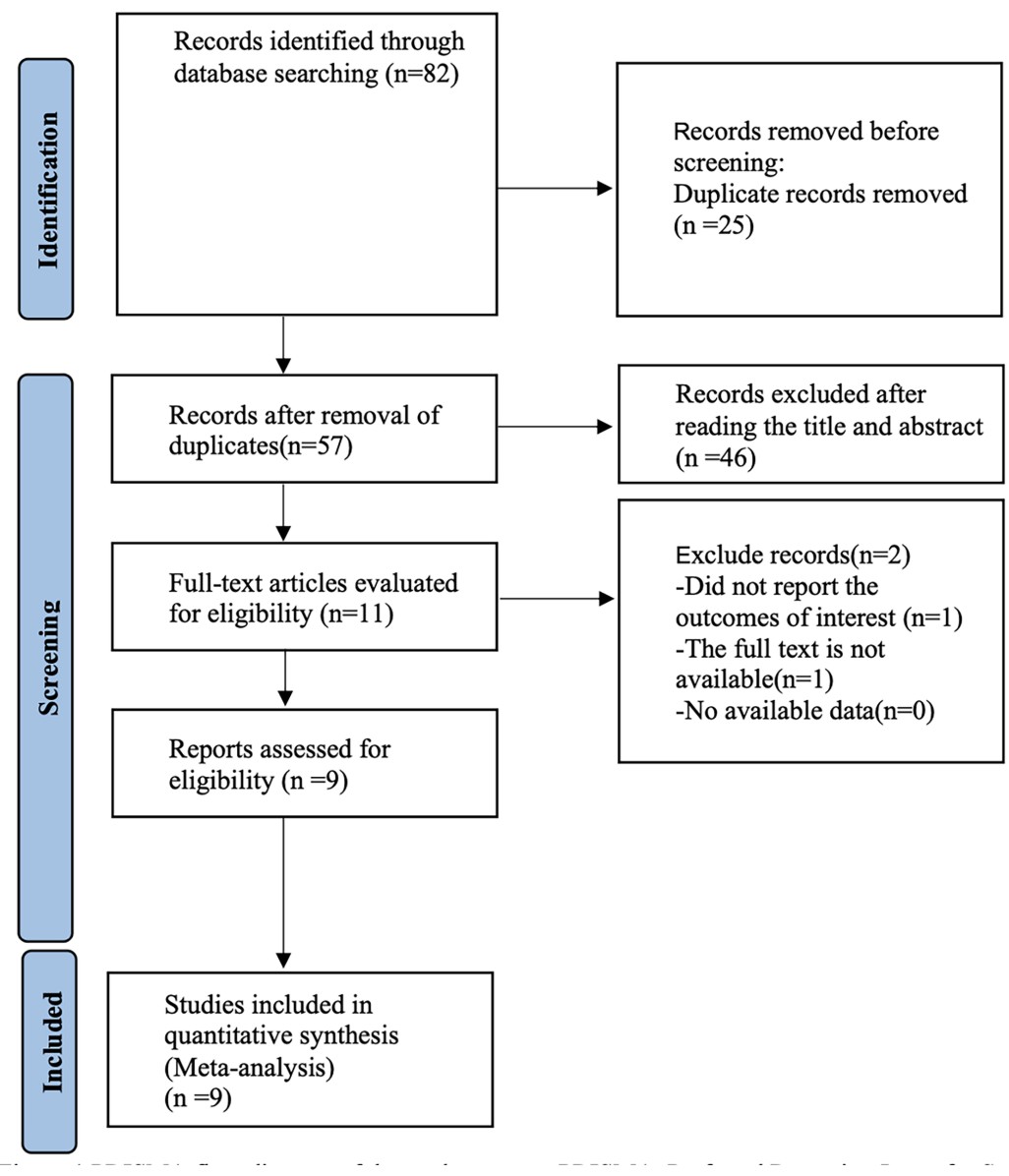

**Figure 1** PRISMA flowchart.

across different time points, the combined analysis consistently demonstrated that EPSB is moderately effective and stable in reducing postoperative pain.

### PCA (analgesia medication use)

Seven articles mentioned PCA (analgesia medication use), heterogeneity test ($I^2 = 95.5\%$, $P = 0.001$), and combined analysis using a random effects model, the results of the analysis (Fig. 7) suggested that EPSB was able to reduce the PCA (analgesia medication use) after spinal fusion (SMD $= -1.67$, 95% CI [$-2.67$ to $-0.67$], GRADE: Moderate). Sensitivity

**Table 1   Table of basic characteristics of the literature.**

| Author | Study | Country | Sample size | | Gender(M/F) | Mean age | | Intervention | | Outcome |
|---|---|---|---|---|---|---|---|---|---|---|
| | | | EG | CG | | EG | CG | EG | CG | |
| Bellantonio | 2023 | Italy | 15 | 15 | 15/15 | 54.6 | 60.4 | ESPB: peripheral intravenous catheter 2% propofol 2 mg/kg | Control: saline solution | F1; F3; F4; F5 |
| Beltrame | 2023 | Argentina | 20 | 20 | 23/17 | 55.6 | | ESPB: Bilateral Radioscopically Guided 0.5% bupivacaine 5 ml | Control: saline solution | F4; F5 |
| Gişi | 2023 | Turkey | 21 | 21 | 21/21 | 51.9 | 50.71 | ESPB: ultrasound guided 2–3 mg/kg propofol | Control: saline solution | F1; F4 |
| Goel | 2021 | India | 51 | 50 | 42/59 | 52.42 | 52.16 | ESPB: ultrasound guided 2–3 mg/kg propofol | Control: saline olution | F1; F3; F4 |
| Kumar | 2024 | India | 28 | 28 | 24/32 | 48.29 | 50.29 | ESPB: ultrasound guided 2% propofol 2 mg/kg | Control: saline solution | F4 |
| Nashibi | 2023 | Iran | 35 | 35 | 32/33 | 50.06 | 52.13 | ESPB: ultrasound guided midazolam 0.02 mg/kg | Control: saline solution | F1; F5 |
| Wang | 2021 | China | 102 | 102 | 95/109 | 53.78 | 55.69 | ESPB: ultrasound guided 2–3 mg/kg propofol | Control: saline solution | F1; F5 |
| Zhang | 2021 | China | 30 | 30 | 15/45 | 60 | 59.5 | ESPB: Bilateral ultrasound guided midazolam 0.02 mg/kg | Control: saline solution | F1; F3; F5 |
| Zhang | 2023 | Chian | 30 | 30 | 24/36 | 61 | 60 | ESPB: Bilateral ultrasound guided midazolam 0.02 mg/kg | Control: saline solution | F1; F3; F4; F5 |

Notes.

EG: Experimental group; CG: Control group; M/F: Male/female; F1: PCA (sedation medication use); F3: Intraoperative blood loss; F4: pain scores; F5: length of hospital stays.

*Bellantonio et al., 2023; Beltrame, Fasano & Jalón, 2023; Gişi & Öksüz, 2023; Goel et al., 2021; Kumar et al., 2024; Nashibi et al., 2023; Wang et al., 2021; Zhang et al., 2021; Zhang et al., 2023.*

analysis was performed using one-by-one exclusion, and the results of the analysis suggested that there was less heterogeneity in the studies and that the analysis was stable.

### *Intraoperative blood loss*

Four articles mentioned intraoperative blood loss, heterogeneity test ($I^2 = 87.7\%$, $P = 0.001$), and combined analysis using a random effects model, the results of the analysis (Fig. 8) suggested that EPSB was does not affect intraoperative blood loss after spinal fusion (SMD $= -0.28$, 95% CI $[-1.03–0.47]$), GRADE: Low). Sensitivity analysis was performed using one-by-one exclusion, and the results of the analysis suggested that there was less heterogeneity in the studies and that the analysis was stable.

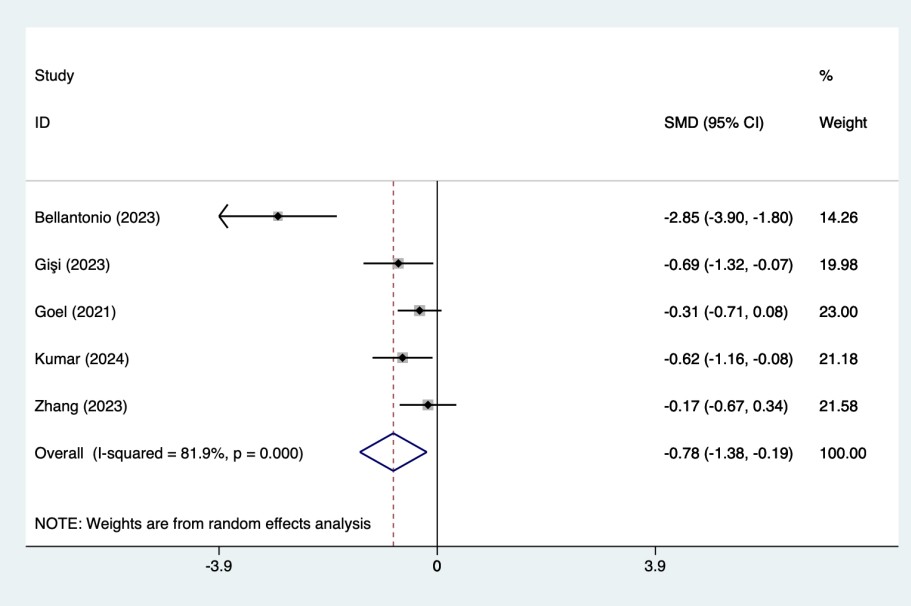

**Figure 2** Forest map of 2 h pain scores meta-analysis (squares represent effect estimates for individual studies, lines represent their confidence intervals; the center of the diamond represents the pooled effect estimate, and the two ends indicate their 95% confidence intervals. Vertical line: indicates the position of "null" or "zero effect" and is used to judge the statistical significance of the effect). *Bellantonio et al., 2023*; *Gişi & Öksüz, 2023*; *Goel et al., 2021*; *Kumar et al., 2024*; *Nashibi et al., 2023*; *Zhang et al., 2023*.

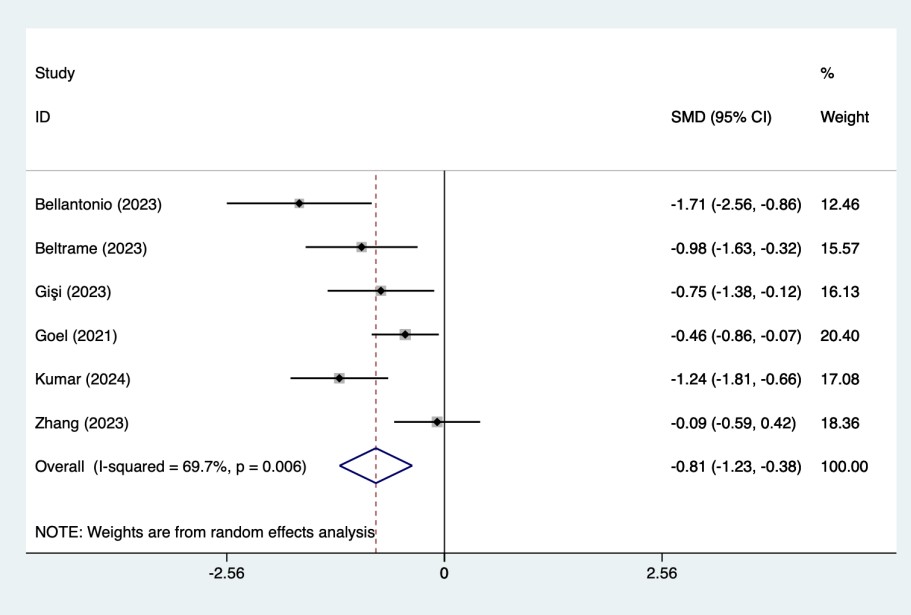

**Figure 3** Forest map of 6 h pain scores meta-analysis (squares represent effect estimates for individual studies, lines represent their confidence intervals; the center of the diamond represents the pooled effect estimate, and the two ends indicate their 95% confidence intervals. Vertical line: indicates the position of "null" or "zero effect" and is used to judge the statistical significance of the effect). *Bellantonio et al., 2023*; *Beltrame, Fasano & Jalón, 2023*; *Gişi & Öksüz, 2023*; *Goel et al., 2021*; *Kumar et al., 2024*; *Nashibi et al., 2023*; *Zhang et al., 2023*.

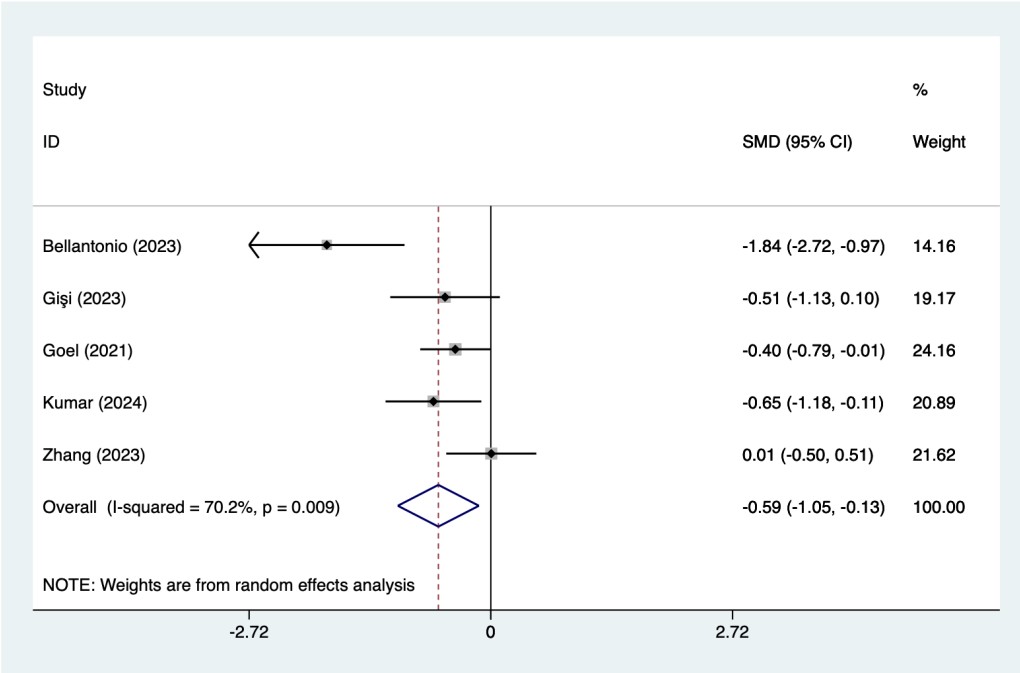

**Figure 4** Forest map of 12 h pain scores meta-analysis (squares represent effect estimates for individual studies, lines represent their confidence intervals; the center of the diamond represents the pooled effect estimate, and the two ends indicate their 95% confidence intervals. Vertical line: indicates the position of "null" or "zero effect" and is used to judge the statistical significance of the effect). *Bellantonio et al., 2023; Beltrame, Fasano & Jalón, 2023; Gişi & Öksüz, 2023; Goel et al., 2021; Kumar et al., 2024; Zhang et al., 2023.*

### Length of hospital stays

Four articles mentioned length of hospital stays, heterogeneity test ($I^2 = 62.8\%$, $P = 0.019$), and combined analysis using a random effects model, the results of the analysis (Fig. 9) suggested that EPSB was does not affect length of hospital stays after spinal fusion (SMD = $-0.27$, 95% CI [$-0.60$–$0.06$], GRADE: Low). Sensitivity analysis was performed using one-by-one exclusion, and the results of the analysis suggested that there was less heterogeneity in the studies and that the analysis was stable.

## Publication bias

In the current study, the Egger test was used to detect publication bias in the included studies, and the results of the study found that no significant publication bias was detected for each of the endpoints ($P > 0.05$).

## DISCUSSION

A previous article (*Viderman et al., 2022*) mentioned the impact of EPSB on surgery in the lumbar spine, and it was found that EPSB was able to reduce postoperative pain and patient satisfaction was high. To the best of our knowledge, this is the first time that meta-analysis was used to evaluate the effect of EPSB on spinal fusion surgery, and our study found that

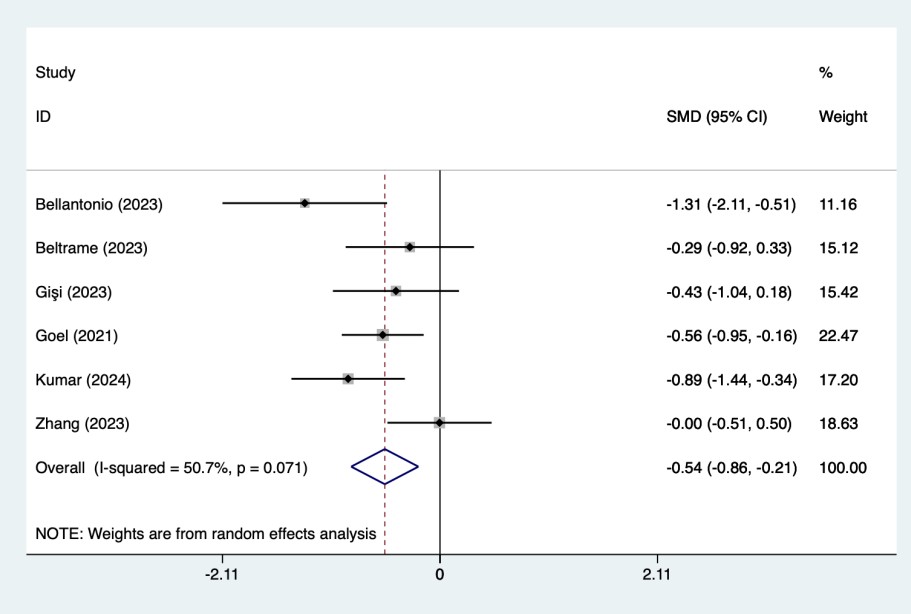

**Figure 5** Forest map of 24 h pain scores meta-analysis (squares represent effect estimates for individual studies, lines represent their confidence intervals; the center of the diamond represents the pooled effect estimate, and the two ends indicate their 95% confidence intervals. Vertical line: indicates the position of "null" or "zero effect" and is used to judge the statistical significance of the effect). *Bellantonio et al., 2023; Beltrame, Fasano & Jalón, 2023; Gişi & Öksüz, 2023; Goel et al., 2021; Kumar et al., 2024; Zhang et al., 2023.*

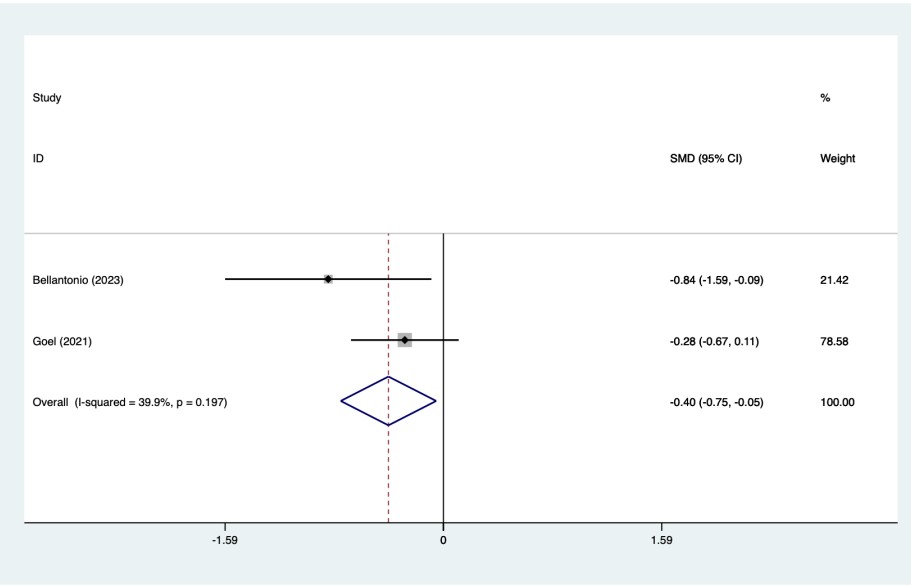

**Figure 6** Forest map of 48 h pain scores meta-analysis (squares represent effect estimates for individual studies, lines represent their confidence intervals; the center of the diamond represents the pooled effect estimate, and the two ends indicate their 95% confidence intervals. Vertical line: indicates the position of "null" or "zero effect" and is used to judge the statistical significance of the effect). *Bellantonio et al., 2023; Goel et al., 2021.*

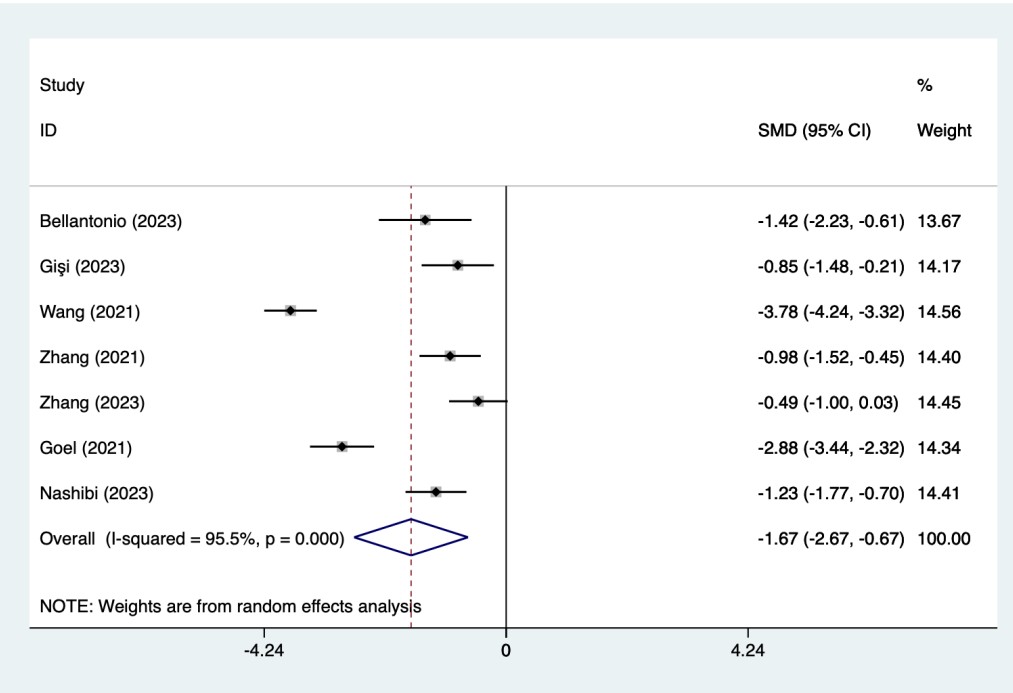

**Figure 7 Forest map of PCA (analgesia medication use) meta-analysis (squares represent effect estimates for individual studies, lines represent their confidence intervals; the center of the diamond represents the pooled effect estimate, and the two ends indicate their 95% confidence intervals.** Vertical line: indicates the position of "null" or "zero effect" and is used to judge the statistical significance of the effect). *Bellantonio et al., 2023*; *Gişi & Öksüz, 2023*; *Goel et al., 2021*; *Nashibi et al., 2023*; *Wang et al., 2021*; *Zhang et al., 2021*; *Zhang et al., 2023*.

EPSB was able to reduce postoperative pain after spinal fusion surgery and was able to reduce the use of analgesics, but it had no effect on intraoperative blood loss or length of hospital stay.

This study found that EPSB was able to reduce postoperative pain after spinal fusion surgery as postoperative time increased, the analgesic effect gradually decreased, possibly due to the half-life of the drugs used in ESPB. The results have been consistent with the results of the meta-analysis by *Cai et al. (2020)* that ESP block significantly reduced patients' pain scores at 1, 6, 12, and 24 h of postoperative rest or exercise, and that ESP block, as a novel technique with good postoperative analgesia, may reduce complications in the early postoperative period of spine, thoracic, and abdominal surgeries. However, regardless of the time change, EPSB could reduce pain scores and analgesics use. This is consistent with the findings of *Singh et al. (2020)*, who observed that in lumbar decompression surgery, patients receiving ESPB had significantly lower postoperative morphine consumption compared to the control group ((1.4 ± 1.5) mg *vs* (7.2 ± 2.0) mg), and immediate postoperative pain scores and pain scores at 6 h postoperatively were also significantly lower in the ESPB group compared to the control group; patient satisfaction scores in the ESPB group were significantly higher than in the control group. *Restrepo-Garces et al. (2017)* demonstrated

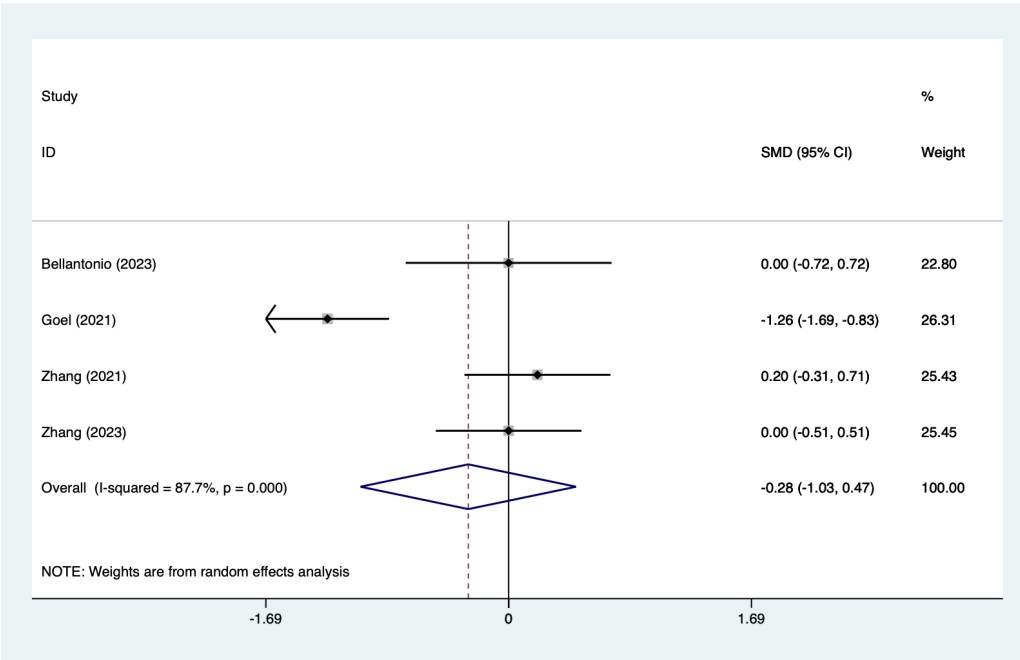

**Figure 8** Forest plot of intraoperative blood loss meta-analysis (squares represent effect estimates for individual studies, lines represent their confidence intervals; the center of the diamond represents the pooled effect estimate, and the two ends indicate their 95% confidence intervals. Vertical line: indicates the position of "null" or "zero effect" and is used to judge the statistical significance of the effect). *Bellantonio et al., 2023*; *Goel et al., 2021*; *Zhang et al., 2021*; *Zhang et al., 2023*.

that in patients undergoing lumbar fusion surgery, both resting and active pain scores were significantly lower in the ESPB group at 4 h postoperatively compared to the control group; fewer patients in the ESPB group required fentanyl within 12 h postoperatively, and the amount of fentanyl used was also significantly reduced, but because analgesia is administered and dosages are inconsistent, these potential confounding factors could affect our results. These outcomes should therefore be interpreted with caution. *Yayik et al. (2019)* found that after lumbar decompression surgery, both resting and active VAS pain scores were lower in the ESPB group compared to the control group, and the time to first analgesic requirement was significantly longer in the ESPB group compared to the control group. After exiting the intervertebral foramen, the spinal nerves divide into ventral branches, dorsal branches, and communicating branches. The ventral branches run horizontally to form the intercostal nerves, initially running deep to the internal intercostal membrane, then between the internal and innermost intercostal muscles, and finally continuing as the anterior cutaneous branches supplying the anterior chest wall and upper abdomen, with lateral cutaneous branches branching off near the costal angle to supply the lateral chest wall (*Elsharydah et al., 2023*; *Jiang et al., 2023*). The dorsal branches pass through the intertransverse ligaments and supply the erector spinae muscles, branching into lateral and intermediate branches, with the intermediate branches ultimately giving rise to the posterior cutaneous branches. The vertebral bodies and paraspinal muscles are

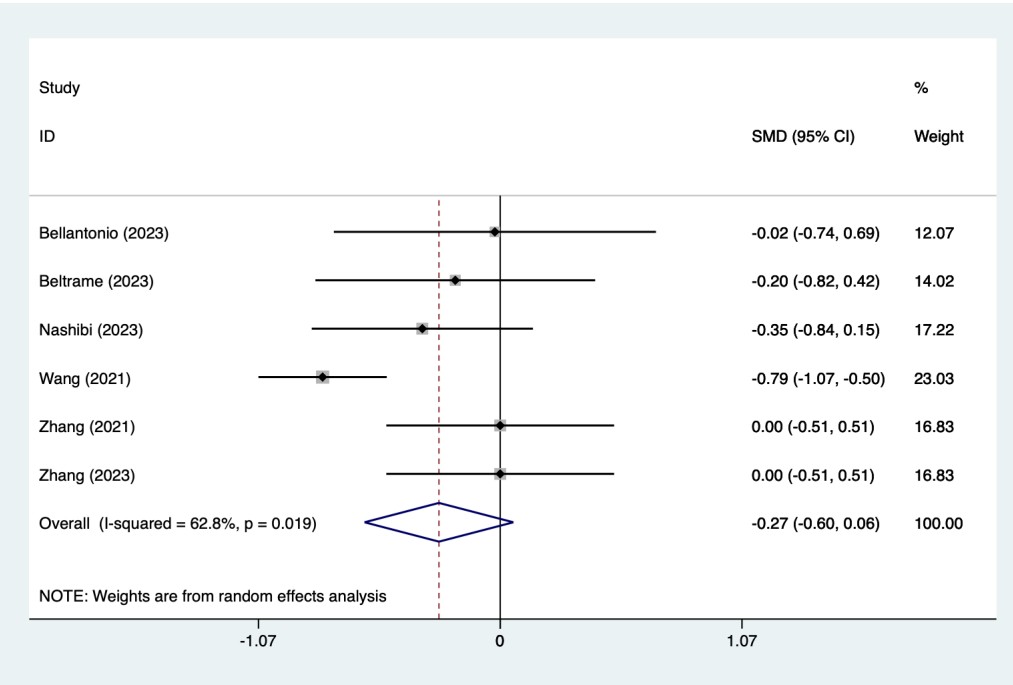

**Figure 9** **Forest plot of length of hospital stays meta-analysis (squares represent effect estimates for individual studies, lines represent their confidence intervals; the center of the diamond represents the pooled effect estimate, and the two ends indicate their 95% confidence intervals.** Vertical line: indicates the position of "null" or "zero effect" and is used to judge the statistical significance of the effect). *Bellantonio et al., 2023*; *Beltrame, Fasano & Jalón, 2023*; *Nashibi et al., 2023*; *Wang et al., 2021*; *Zhang et al., 2021*; *Zhang et al., 2023*.

innervated by the dorsal branches of the spinal nerves. ESPB can block not only the ventral branches of the spinal nerves but also the dorsal branches, which is the anatomical basis for its analgesic effect (*Sharma et al., 2023*).

As a new nerve block technique, ESPB is highly safe, and compared with traditional intravertebral block and thoracic paravertebral block, the injection point is superficial and not close to important organs and blood vessels, so the risk of complications such as pneumothorax, hematoma, and neurological injuries is lower, and the requirements for coagulation are also lower, so it is easy to be promoted in clinical practice (*De Cassai, Stefani & Ori, 2018*). This study still has the following limitations: first, the type and quantity of postoperative PCA analgesics are inconsistent, which may be a source of clinical heterogeneity; second, the small number of studies included in this study and the small number of researchers involved may affect the credibility of the conclusions of this study; third, the inclusion of the population, the use of systemic analgesia and the use of nerve block drugs in the literature are not completely consistent, which may be a source of clinical heterogeneity.

## CONCLUSION

Combined with the current findings, EPSB may reduce pain scores in spinal fusion surgery, possibly reducing the use of postoperative analgesics. However, due to the limitations of the study, we need more high-quality, multi-center, large-sample randomized controlled trials to merge.

### Funding
The authors received no funding for this work.

### Competing Interests
The authors declare there are no competing interests.

### Author Contributions
- Yi He conceived and designed the experiments, performed the experiments, analyzed the data, prepared figures and/or tables, authored or reviewed drafts of the article, and approved the final draft.
- Heng Liu conceived and designed the experiments, performed the experiments, analyzed the data, prepared figures and/or tables, authored or reviewed drafts of the article, and approved the final draft.
- Peng Ma conceived and designed the experiments, performed the experiments, analyzed the data, prepared figures and/or tables, authored or reviewed drafts of the article, and approved the final draft.
- Jing Zhang conceived and designed the experiments, performed the experiments, analyzed the data, prepared figures and/or tables, authored or reviewed drafts of the article, and approved the final draft.
- Qiulian He conceived and designed the experiments, performed the experiments, analyzed the data, prepared figures and/or tables, authored or reviewed drafts of the article, and approved the final draft.

### Data Availability
This is a systematic review/meta-analysis.

### Supplemental Information
Supplemental information for this article can be found online at http://dx.doi.org/10.7717/peerj.18332#supplemental-information.

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
