# Peer review of "Meta-analysis of the efficacy of the erector spinae plane block after spinal fusion surgery"

_PeerJ, doi:10.7717/peerj.18332_

## Round 0.1 · original submission · Major Revisions

Thank you for submitting this manuscript for review. The reviewers have all provided valuable comments, and in your resubmission you should respond to every concern of the reviewers with a response that includes what changes you have made in response. It will certainly be important to explain any deviations from the PROSPERO protocol that you submitted. If you believe that all the figures are essential, then you are welcome to retain the in the manuscript.

Reviewer 1 ·

Basic reporting

I have reviewed the abstract, introduction, methods and materials, results, statistics, and discussion. I have also checked the references, and all appear relatively current and appropriate. Finally, I have also reviewed the figures, tables, and legends.
I appreciate the clarity and complexity of the presentation.
However, I have a few suggestions aiming to improve the manuscript quality:
- I have found the discrepancies between the protocol registered in PROSPERO and the protocol described in the manuscript:
- in the manuscript, you wrote that you searched the Embase, but in the PROSPERO protocol, you wanted to search only PubMed, Cochrane, and Web of Science. Please clarify



- in the discussion section- you did not discuss your outcomes. You should rewrite your discussion and compare your results to other meta-analyses regarding ESPB

Experimental design

What was our PICO question? You should add it to the manuscript.

The inclusion and exclusion criteria differ from the protocol registered in PROSPERO and the one described in the manuscript. Adhering to the protocol registered in PROSPERO is crucial, as it ensures the research's validity and maintains the scientific community's trust. It would be best to describe the inclusion and exclusion criteria in detail so they can be observed.

Can you provide the Figure for the risk of bias, showing which studies showed a risk of bias? It would improve the clarity of the presentation and would be beneficial for the readers.
Also, I suggest that you limit your analysis to trials with a low risk of bias.

I suggest adding a secondary scale to evaluate the risk of bias. While you used the Cochrane Risk of Bias Assessment tool, it would provide a more comprehensive evaluation of the study's quality and assess the Jadad score.

Validity of the findings

Please be specific about the GRADE evidence and provide a detailed table of the GRADE evaluation. For each outcome, give the number of studies, study design, quality of assessment (risk of bias, inconsistency, imprecision, other), number of patients in the control and the espb, the effect of the outcome, and in the last column, the quality.

Additional comments

- in the discussion section- you did not discuss your outcomes. You should rewrite your discussion and compare your results to other meta-analyses regarding ESPB

Reviewer 2 ·

Basic reporting

The article is written in clear, unambiguous, and technically correct English throughout.

The title of the manuscript does not accurately describe the contents of the manuscript.
• The title describes the efficacy of Erector Spinae Plane block for spinal fusion surgery”. Still, the manuscript did not mentions the impact of ESPB on intra-operative opioid consumption. If the author wants to focus on postoperative pain, it is advisable to add “postoperative “in the title.

The abstract was written well, according to IMRAD principles.
• However, the objective stated in the abstract does not align with the title. The “safety” of ESPB is absent from the objective stated in the abstract

The literature references are extensive and relevant, demonstrating a thorough understanding of prior research and how this meta-analysis contributes to existing knowledge.

However, the introduction is not adequately describes the background of the systematic review.
• Author must elucidate the reasons why spinal fusion surgery is distinct from other spine surgeries in order to conduct an analysis, since metaanalysis of ESPB in apine surgery had been published previously.
• The usage of the phrase "sedative medication" in this manuscript is incorrect. PCA typically employs morphine or other opioids, which are categorized as analgesics rather than sedatives. Pain management in the post-operative period of spinal fusion surgery does not involve the use of sedative drugs

The manuscript follows an acceptable format of standard sections.
The figures and tables included are relevant and appropriately described and labeled, significantly aiding in the presentation of data and findings.

Experimental design

The authors have performed an extensive literature search using specific terms and suitable constraints. The issue of data selection bias was successfully addressed, and two interdependent writers utilized a structured data extraction table. The inclusion and exclusion criteria were clearly defined and objective, and the quality of the selected randomized controlled studies was appraised using an appropriate technique.The chosen randomized controlled trials (RCTs) are compatible and subjected to statistical analysis utilizing reliable and robust methodologies. Sensitivity testing are appropriately employed.

Validity of the findings

The results section of this the text provides a detailed description of the pain score at 6, 12, 24, and 48 hours, as well as the amount of blood loss. However, the wording used to explain these measures is redundant. The author must articulate these facts better and in more efficient way.

In the discussion chapter, the author does not effectively illustrate the disagreement between research findings and the factors that influence them.

The conclusions of the manuscript do not align with the title and obejcetives of the meta-analysis.

Additional comments

Overall, this meta-analysis is a valuable contribution to the field, combining rigorous methodology with clear presentation and thorough contextualization within the existing literatur.
This meta analysis covers a topic that is currently widely discussed. Several publications similar to this meta-analysis have been published. Unfortunately, the author fail to show in his manuscript the differences between this manuscript and various existing publications. The choice of "Spinal Fusion Surgery" as a differentiating factor from other publications cannot be explained well, either in the introduction chapter or in the discussion. In fact, this differentiating factor is precisely the strength of this manuscript

Reviewer 3 ·

Basic reporting

The manuscript should be refined to make it clear and coherent. It was wordy to repeat the same meaning with almost the same sentences (Lines 56, 57). Some expressions should be modified, such as “evidence-based basis” (Line 56, 57), “use of opioid analgesics. . . is associated with a higher dose” (Line 37, 38), “secondary outcome metrics” (Line 69), “The present study was mainly included in randomized controlled studies for analysis” (Line 70, 71), (Line 118), et al.
The authors confused anesthesia and analgesia. In the abstract (Line 24, 29), PCA is the method for analgesia but not anesthesia or sedation. In the introduction (Line 56, 57), EPSB is for analgesia but not anesthesia. The mistake happened in the text too.
Too many pictures. I failed to find any figure legends. The style of the table needed to be refined.
The flowchart was Figure 1, which was mistakenly identified as SFig.1 (Line 116).
Abbreviations should be defined (such as SMD).
Line 105 “I2” lost its superscript style.

Experimental design

The type of surgery should be accurate. The authors stated “spinal fusion surgery” in the title, but used “traditional spinal surgeries” and “spinal surgery” instead of “spinal fusion surgery” throughout the introduction. If the hypothesis was based on “traditional spinal surgeries”, it may not be suitable for “spinal fusion surgery”.
The Grade of evidence needs a more detailed description regarding the criteria for declaring “moderate” or “mild”.
Different technique details regarding ESPB from the studies should be disclosed, such as the dose and type of the local anesthetics. This information is essential for readers to assess the anticipated effect of the ESPB.
The pain score needed a clear definition. VAS? NRS? CPOT? Different pain scores may have different ranges. How did the authors normalize them?
The authors need to explain the details regarding the measure of the PCA. Analgesic drugs used in PCA (not sedation medication which was used in the manuscript by the authors) vary among different centers. How did the authors normalize them?
I failed to find how many participants had been included in this study.

Validity of the findings

The introduction was not persuasive. A logical statement is needed for the readers to comprehend the aim of this study. The discussion did not give too much impressive information.
The conclusion “EPSB was found to . . . decrease the use of postoperative sedative medications” was not right. Sedative medications are not used postoperatively.

Additional comments

Thanks to the authors for their work regarding the acute pain control effect of ESPB. Since side effects were not assessed, using “safety” in the title may not be suitable.
Language polishing will help this manuscript be clear and intelligible.
Considering some important details and definitions should be stated, authors may be recommended to refine this manuscript further.

---

## Round 0.2 · Major Revisions

Reviewer 3 continues to have significant concerns about the content of your manuscript, and has kindly provided additional guidance to you to help improve the manuscript. In your rebuttal, please address each comment, provide your response, and indicate how the manuscript has been changed.

Reviewer 2 ·

Basic reporting

No comment

Experimental design

No comment

Validity of the findings

No comment

Additional comments

The manuscript has been well written and has met the requirements for publication.

Reviewer 3 ·

Basic reporting

I want to extend my sincere thanks to the authors for their diligent work on revising the manuscript. This version is indeed a significant improvement over the previous one. However, there seems still some concerns that remain unaddressed.

Major Concerns:

1. I must emphasize the importance of including patient-controlled analgesia (PCA) in a study focused on postoperative pain management. PCA is a standard and essential method for managing postoperative pain, and its omission makes the assessment of the pain management approach less persuasive.

2. I respectfully disagree with the notion that different pain scores can be simply normalized and combined into a single unified measurement by using difference values. Each pain score is applied in different scenarios and has distinct ranges (e.g., VAS: 0-100, NRS: 0-10, CPOT: 0-8). It is not rational to treat their difference values as equivalent.

3. I recommended disclosing the different technique details regarding ESPB from the studies. Instead of providing this information, the authors added sedatives and local anesthetics to Table 1, which only added to my confusion. General anesthetics cannot be considered part of a fascial plane block technique. A more appropriate response would involve detailing whether a single-injection block was used or if a catheter was placed for continuous relief, as well as whether the procedure was performed with ultrasound guidance, among other relevant technique details.

4. My comments regarding the assessment of the grade of evidence in this study were not addressed. The GRADE system evaluates study design, quality, consistency, and directness. The authors stated that they used the GRADE system but listed five factors they considered, including study limitations, inconsistent findings, inconclusive direct evidence, imprecise or wide confidence intervals, and publication bias, along with three additional factors: effect size, possible confounding factors, and dose-effect relationships. Given these discrepancies, it is essential to ensure that readers clearly understand the methodology used to determine the quality of evidence in this study.

Minor Concerns:

1. I recommend that the authors carefully read through the entire manuscript before submission. Even in the revised sections, I noticed a clerical error with an extra "s" (Line 62). There may be additional issues that could be identified with a more thorough review.

2. While figure titles have been added to address my previous request, I want to clarify that a figure legend is generally not the same as a figure title. Some figures would benefit from a brief legend to explain abbreviations and provide additional context.

3. The conclusion still overstates the findings by claiming that EPSB decreases the use of postoperative medication based on this study. This assertion needs to be toned down to accurately reflect the study's results.

Experimental design

no comment

Validity of the findings

no comment

Additional comments

no comment

---

## Round 0.3 · accepted · Accept

Thank you for being responsive to the last set of reviews. I believe you have now addressed all comments satisfactorily, and am pleased to accept your manuscript for publication.

Reviewer 1 ·

Basic reporting

I have reviewed the abstract, introduction, methods and materials, results, statistics, and discussion. I have also checked the references, and all appear relatively current and appropriate. Finally, I have also reviewed the figures, tables, and legends.
I appreciate the clarity and complexity of the presentation.

Experimental design

I have reviewed the abstract, introduction, methods and materials, results, statistics, and discussion. I have also checked the references, and all appear relatively current and appropriate. Finally, I have also reviewed the figures, tables, and legends.
I appreciate the clarity and complexity of the presentation.

Validity of the findings

I have reviewed the abstract, introduction, methods and materials, results, statistics, and discussion. I have also checked the references, and all appear relatively current and appropriate. Finally, I have also reviewed the figures, tables, and legends.
I appreciate the clarity and complexity of the presentation.